# Virulence and Antimicrobial Resistance Profiles of *Salmonella enterica* Serovars Isolated from Chicken at Wet Markets in Dhaka, Bangladesh

**DOI:** 10.3390/microorganisms9050952

**Published:** 2021-04-28

**Authors:** Nure Alam Siddiky, Md Samun Sarker, Md. Shahidur Rahman Khan, Ruhena Begum, Md. Ehsanul Kabir, Md. Rezaul Karim, Md. Tanvir Rahman, Asheak Mahmud, Mohammed A. Samad

**Affiliations:** 1Antimicrobial Resistance Action Center, Bangladesh Livestock Research Institute, Savar, Dhaka 1341, Bangladesh; nasiddiky.saarc@gmail.com (N.A.S.); samuncvasu@gmail.com (M.S.S.); dr.ruhenabegum@gmail.com (R.B.); tanjib_bau@yahoo.com (M.E.K.); rezavetmicro@yahoo.com (M.R.K.); jnumicroasheak@gmail.com (A.M.); 2Department of Microbiology and Hygiene, Bangladesh Agricultural University, Mymensingh 2202, Bangladesh; msrkhan001@yahoo.com (M.S.R.K.); tanvirahman@bau.edu.bd (M.T.R.); 3Faculty of Veterinary Medicine, Universiti Putra Malaysia, Serdang 43400, Selangor, Malaysia

**Keywords:** *Salmonella**enterica* serovars, cecal contents, virulence, AMR, wet markets

## Abstract

Virulent and multi drug resistant (MDR) *Salmonella*
*enterica* is a foremost cause of foodborne diseases and had serious public health concern globally. The present study was undertaken to identify the pathogenicity and antimicrobial resistance (AMR) profiles of *Salmonella*
*enterica* serovars recovered from chicken at wet markets in Dhaka, Bangladesh. A total of 870 cecal contents of broiler, sonali, and native chickens were collected from 29 wet markets. The overall prevalence of *S.* Typhimurium, *S.* Enteritidis, and untyped *Salmonella* spp., were found to be 3.67%, 0.57%, and 1.95% respectively. All isolates were screened by polymerase chain reaction (PCR) for eight virulence genes, namely *inv*A, *agf*A, *Ipf*A, *hil*A, *siv*H, *sef*A, *sop*E, and *spv*C. *S*. Enteritidis isolates carried all virulence genes whilst *S*. Typhimurium isolates carried six virulence genes except *sef*A and *spv*C. A diverse phenotypic and genotypic AMR pattern was found. Harmonic descending trends of resistance patterns were observed among the broiler, sonali, and native chickens. Interestingly, virulent and MDR *Salmonella enterica* serovars were found in native chicken, although antimicrobials were not used in their production cycle. The research findings anticipate that virulent and MDR *Salmonella enterica* are roaming in the wet markets which can easily anchor to the vendor, consumers, and in the food chain.

## 1. Introduction

*Salmonella* is a vital food-borne infection in humans across the world and had significant morbidity, mortality, and economic loss [1,2]. *Salmonella* is a prominent driver of foodborne illness worldwide with an anticipated annual economic burden 3.7 billion dollars [3]. *Salmonella* is one of the most common foodborne pathogens, causing outbreaks of foodborne disease globally [4]. Poultry in particular have been regarded as the single prime cause of human salmonellosis although the pathogen have been associated with a diverse variety of food sources. Avian salmonellosis is not only affects the poultry industry but also can infect humans and caused by consumption of contaminated poultry meat and eggs [5]. Livestock products and by-products especially eggs and poultry meats are the common carriers of *Salmonella* infections [6]. Foodborne *Salmonella* infection may cause typhoid and enteritis and may have more severe in the immune-depressed people [7]. A diverse prevalence of *Salmonella* has been reported from poultry worldwide. The global trend of *Salmonella* infection has been intensified due to increasing consumption of livestock and poultry products [7,8]. Several researchers reported variable prevalence rates of *Salmonella* infection in animal food sources in Bangladesh [9,10]. *Salmonella enterica* serovars Typhimurium (*S*. Typhimurium) and Enteritidis (*S*. Enteritidis) are the most common serovars linked with human foodborne illness among the more than 2500 serovars [11,12]. Human *S*. Enteritidis cases are mostly associated with the consumption of contaminated eggs and poultry meat, while *S*. Typhimurium cases with the consumption of contaminated pork, poultry, and beef [13,14]. A diverse prevalence of *Salmonella enterica* serovars has been reported in many countries across the globe among the animal products and by products [12,15,16]. It is universal fact that human *Salmonella* illness are related with numerous diverse kinds of food, including animal origin food and food products [2,17]. In Bangladesh, limited information has been published on *Salmonella enterica* serovars isolated in chicken meat or from chicken cecal contents at wet markets.

Generally the virulence factors trigger the pathogenicity of *Salmonella* infection. The virulence of *Salmonella* is interlined with combination of chromosomal and plasmid factors. Diverse genes, such as *inv*A, *agf*A, *Ipf*A, *hil*A, *siv*H, *sef*A, *sop*E, and *spv*C, have been recognized as major virulence genes liable for salmonellosis. The infectivity of *Salmonella* strains is associated with various virulence genes existent in the chromosomal *Salmonella* pathogenicity islands (SPIs) [18]. The invasion genes *inv*A, *hil*A, and *siv*H code with a protein in the inner chromosomal membrane of *Salmonella* that is necessary for the invasion to epithelial cells [19]. Moreover, *Salmonella* effector protein adhered by *sop*E gene which have relevance to *Salmonella* virulence [20]. The plasmid mediated *spv*C gene is responsible for vertical transmission [21]. The long polar fimbria (*Ipf* operon) make the attraction of the microbes for Peyer’s patches and adhesion to intestinal M cells [22]. The aggregative fimbria (*agf* operon) promote the primary interaction of the *Salmonella* with the intestine of the host and stimulate microbial self-aggregation for higher rates of survival [23]. The *Salmonella*-encoded fimbria (*sef* operon) endorses interaction between the microbes and the macrophages [23]. There is a very inadequate information exist in the determination of virulence gene from *Salmonella enterica* serovars in Bangladesh.

Antibiotics are one of the most powerful tools for fighting life-threatening infections. Their discovery has transformed human and animal health. Unfortunately, we now live in an era when people around the world are dying from untreatable infections because of the emergence and spread of AMR. More than 2.8 million antibiotic-resistant infections occur in the United States each year, and more than 35,000 people die as a result [24]. AMR poses a formidable challenge to achieving sustainable development goals, including in health, food security, clean water and sanitation, responsible consumption and production, and poverty and inequality. Misuse and overuse of existing antimicrobials in humans, animals and plants are accelerating the development and spread of AMR [25]. The trends of MDR *Salmonella* strains has been increasing on a worldwide scale, especially in the food animals [26]. In Bangladesh, the antimicrobials are used as the therapeutic, preventive and growth prompters in the animal production system [27]. It is evident that antimicrobials are used as growth promoters in poultry [28]. The burden of AMR *Salmonella* has become a worldwide concern in recent decade [29,30]. MDR *Salmonella* of animal origin have been increasing in Bangladesh. The prevalence and resistance of *Salmonella* in either animal [31,32,33,34], or chicken farms or in retail poultry meats of wet market [34], or in slaughtering processes [35,36] have been detected. Moreover, several studies have been carried out at the molecular level to monitor the distribution of resistance genes in *Salmonella* serovars isolated from broiler chickens and chicken meat [37,38,39]. The dissemination of phenotypic and genotypic resistance gene in the food production chain and slaughtering process has been identified by different researchers [40,41,42].

Wet markets are very common in most South Asian countries, including Bangladesh. The vendors in wet markets are usually dress the chicken himself and carcass wastages are not managed well. Due to poor hygienic and sanitary practices there is a chance of spread and contamination of foodborne pathogen especially *Salmonella*. To the best of our knowledge, it is the first study ever to monitor the virulence gene along with phenotypic and genotypic AMR profiles of *Salmonella enterica* serovars at wet markets in Bangladesh. In our study, we have focused the AMR profiles of *Salmonella enterica* serovars, i.e., *S*. Enteritidis and *S*. Typhimurium due to their public health importance. Moreover, this study is fully aligned with national AMR surveillance strategy of the country. Even most of the national AMR surveillance strategy across the globe suggested to collect cecal samples from healthy chicken at wet markets for AMR surveillance. Therefore, the present study was conducted to identify the prevalence of *S.* Typhimurium and *S*. Enteritidis along with their virulence gene and phenotypic and genotypic AMR properties in chicken at wet markets in Dhaka, Bangladesh.

## 2. Materials and Methods

### 2.1. Study Design and Sample Size

A cross sectional study was conducted at the chicken wet markets in Dhaka city, the capital of Bangladesh from February to December 2019. Dhaka city is considered the biggest chicken selling hub where the chicken was supplied from different destinations of the country. We have selected city corporation authorized 29 out of 165 wet markets having at least 10 chicken vendors in Dhaka city based on random selection technique. It was assumed that daily sells more than 0.5 million chickens at wet markets in Dhaka city. The sample size was calculated based on the prevalence of *Salmonella* spp. found by different previous studies in Bangladesh. Sample size was calculated by using “Sample size calculator for prevalence studies, version 1.0.01” [43,44]. The expected prevalence of *Salmonella* was considered 25% with 95% confidence interval and 5% accepted error precision and poultry population considered 0.5 million during sample size calculation. The sample size calculation showed that the individual type of sample number should not less than 289. Therefore, individual cecal contents of broiler (commercial chicken), sonali (cross breed), and native chickens were collected 290.

This study received ethical approval from the Ethical Committee of the Animal Health Research Division at the Bangladesh Livestock Research Institute, Dhaka, Bangladesh (ARAC: 15/10/2019:04).

### 2.2. Isolation and Identification of Salmonella Spp

Isolation and identification of *Salmonella* were done according to the guidelines of ISO [45] as following; pre-enrichment of the cecal contents in buffered peptone water (BPW; Oxoid, UK) at 1:10 dilution followed by aerobic incubation at 37 °C for 18–24 h. Later, 0.1 mL of the pre-enriched sample was placed separately in three different places on Modified Semisolid Rappaport Vassiliadis (MSRV; Oxoid, UK) agar and incubated at 41.5 °C for 20–24 h. One loop from MSRV was streaked on Xylose Lysine Deoxycholate (XLD; Oxoid, UK) and another loop to MacConkey agar plates and incubated at 37 °C for overnight [46]. Characteristic black centred with reddish zone colony on XLD and colorless colony on MacConkey were picked up and subsequently sub cultured on nutrient agar and subjected to biochemical tests: triple sugar iron (TSI), motility indole urea (MIU), catalase and oxidase. Final confirmation was done by the Vitek-2 compact analyser (bioMérieux, Marcy-l’Étoile, France) followed by PCR.

### 2.3. DNA Extraction

DNA from pure culture was extracted using conventional boiling method [47]. Briefly, each isolate cultured on nutrient agar medium and incubated overnight at 37 °C. Few fresh colonies were harvested from overnight culture and suspended in nuclease free water. Then bacterial suspension was boiled at 99 °C for 15 min followed by chilled on ice. Finally, the debris were separated by centrifugation and supernatant was taken as the DNA template for PCR.

### 2.4. PCR for the Detection of Salmonella Spp and Salmonella enterica Serovars

*Salmonella* was confirmed with the detection of virulence gene *inv*A by an uniplex PCR ((u-PCR-1) with initial denaturation at 95 °C for 1 min; 38 cycles of 95 °C for 30 s, 64 °C for 30 s and 72 °C for 30 s and final elongation at 72 °C for 4 min [48,49]. Multiplex PCR (m-PCR I) was done to detect *S*. Typhimurium and *S*. Enteritidis with initial denaturation at 95 °C for 2 min; 30 cycles of 95 °C for 1 min, 57 °C for 1 min, 72 °C for 2 min, and final elongation at 72 °C for 5 min [50,51,52]. PCR assays were adjusted in 25 µL reaction mixture containing 2 µL of DNA template, 12.5 µL of 2x master mix (Go Taq Green Master Mix, Promega, Dane County, WI, USA), 0.5 μL each of forward and reverse primers (10 pmol/μL) and 9.5 µL nuclease free water. The PCR products were run through 1.5% (*w*/*v*) agarose gel electrophoresis. A 100 bp DNA ladder (Thermo Scientific, USA) was used as a size marker. The primers used in this study for PCR are presented in Appendix A. *S*. Typhimurium ATCC 14028 and *S*. Enteritidis ATCC 13076 were used as positive control in the PCR assay. Consequently, PCR positive *Salmonella* serovars was further reconfirmed by the Vitek-2 compact analyser (bioMérieux, France).

### 2.5. Antimicrobial Susceptibility Test

Antimicrobial susceptibility testing (AST) was made by the Kirby-Bauer disk diffusion method in accordance with the guidelines of the Clinical and Laboratory Standards Institute (CLSI) [53]. Briefly, two-three fresh colonies were suspended in 3 mL normal saline and the turbidity of the suspension was standardized to match with 0.5 McFarland standard (approximately 1.5 × 10^6^ CFU/mL). This bacterial inoculum was wiped over the surface of Mueller Hinton agar (MHA; Oxoid, UK) plate, onto which the antimicrobial disks were placed by using disk dispenser within 15 min. Plates were incubated for 16–24 h at 35–37 °C prior to determination of results. The diameter of the zone of inhibition surrounding the disks was measured by automated zone of inhibition reader (Scan^®^ 4000, Interscience, Paris, France) and compared to the break points of CLSI [53]. The disk diffusion was done against 16 antimicrobials under 10 groups including aminoglycosides: amikacin (AK, 30 µg), gentamicin (CN, 10 µg), streptomycin (S, 10 µg); carbapenem: meropenem (MEM, 10 µg); cephalosporin/beta-lactam antibiotics: ceftriaxone (CRO, 30 µg), cefotaxime (CT, 10 µg), ceftazidime (CAZ, 30 µg), aztreonam (ATM, 30 µg); beta-lactamase inhibitors: amoxicillin–clavulanate (AMC, 30 µg); penicillins: ampicillin (AMP, 10 µg); macrolides: azithromycin (AZM, 15 µg); quinolones/fluoroquinolones: ciprofloxacin (CIP, 5 µg), nalidixic acid (NA, 30 µg); folate pathway inhibitors: sulfamethoxazole-trimethoprim (SXT, 25 µg); tetracycline: tetracycline (TE, 10 µg); phenicols: chloramphenicol (C, 30 µg). *Salmonella* isolates resistant to three or more antimicrobials were defined as MDR isolates [54]. Intermediate was regarded as resistant isolates since the acquisition and transition from susceptible to resistance had already begun [55]. The *Escherichia coli* ATCC 25922 strain was used as known positive control. The multiple antibiotic resistance (MAR) index was calculated and interpreted using the proven method [56,57].

### 2.6. PCR for the Detection of Antimicrobial Resistance Genes

*Salmonella* enterica serovars Typhimurium and Enteritidis were screened by PCR for the detection of *β*-lactamase genes (*bla*TEM, *bla*SHV, *bla*OXA, *bla*CTX-M-1, *bla*CTX-M-2, *bla*CTX-M-9 and *bla*CTX-Mg8/25), tetracycline resistant genes (*tet*A, *tet*B and *tet*C), sulfonamide resistant genes (*sul*1, *sul*2, and *sul*3) and streptomycin resistant gene (*str*A/B). All PCR assays were adjusted in 25 µL reaction mixture containing 2 µL of DNA template, 12.5 µL of 2x master mix (Go Taq Green Master Mix, Promega, Dane County, WI, USA), 0.5 μL each of forward and reverse primers (10 pmol/μL), and 9.5 µL nuclease free water. For *β*-lactam gene, multiplex PCR (m-PCR II and m-PCR III) amplification was carried out with initial denaturation at 94 °C for 10 min; 30 cycles of 94 °C for 40 s, 60 °C for 40 s and 72 °C for 1 min; and a final elongation step at 72 °C for 7 min. Multiplex PCR (m-PCR IV) was conducted to detect sulfonamide resistant gene with initial denaturation at 95 °C for 15 min; 30 cycles of 95 °C for 1 min, 1 min of annealing at 66 °C and 72 °C for 1 min; and a final elongation step at 72 °C for 10 min. Similarly, multiplex PCR (m-PCRV) was done to detect tetracycline and streptomycin resistance gene with initial denaturation at 94 °C for 15 min; 30 cycles of 94 °C for 1 min, 1 min of annealing at 63 °C and 72 °C for 1 min; and a final elongation step at 72 °C for 10 min. The primers used in this study for PCR are presented in Appendix A [50,51,58,59]. Amplicons were visualized after running at 100 V with 500 mA for 30 min in 1.5% agarose gel containing ethidium bromide. A 100-bp DNA ladder (Thermo Scientific, USA) was used as a size marker.

### 2.7. PCR for the Detection of Virulence Genes

All *Salmonella* enterica isolates were screened by PCR to discover virulence genes. The PCRs were executed in single reactions using primers for detection of eight virulence genes *inv*A, *agf*A, *ipf*A, *hil*A, *siv*H, *sef*A, *sop*E, and *spv*C are presented in Appendix A [49,60,61,62,63,64,65,66]. All PCR assays were optimized in 25 µL reaction mixture containing 2 µL of DNA template, 12.5 µL of 2x master mix (Go Taq Green Master Mix, Promega, Dane County, WI, USA), 1 μL each of forward and reverse primers (20 pmol/μL) and 8.5 µL nuclease free water. The reference positive control (*S*. Typhimurium ATCC 14028 and *S*. Enteritidis ATCC 13076) and negative control (*E. coli* ATCC 25922) were used for validation. Amplicons were visualized after running at 100 V with 500 mA for half an hour in 1.2% agarose gel containing ethidium bromide. A 100bp DNA ladder (Thermo Scientific, Waltham, MA, USA) was used as a size marker.

### 2.8. Statistical Analysis

All data were incorporated into Excel sheets (MS-2016) and analyzed by SPSS software (SPSS-20.0). The prevalence was calculated using descriptive analysis and Chi-square test was done to determine the level of significance. Statistical significance was measured by *p*-values less than 0.05 (*p* < 0.05).

## 3. Results

### 3.1. Prevalence of Salmonella enterica Serovars

The isolated *Salmonella* spp., *S*. Typhimurium and *S*. Enteritidis produced PCR products 284 bp, 401 bp, and 293 bp in size, respectively in gel documentation system. The overall prevalence of *Salmonella* spp. was found 8.62% (25 in 290) in broiler, 6.89% (20 in 290) in sonali (cross breed) and 3.1% (nine in 290) in native chicken. Meanwhile, the MDR *Salmonella* spp. was found 84% (21 in 25), 75% (15 in 20), and 44.4% (four in nine) in broiler, sonali, and native chickens, respectively. The prevalence of MDR *S*. Typhimurium and *S*. Enteritidis were found 78.1% and 80%, respectively. Among the 25 *Salmonella* isolates of broiler chicken, the prevalence of *S*. Typhimurium, *S*. Enteritidis and untyped *Salmonella* spp., were found 60% (15 in 25), 20% (five in 25) and 20% (five in 25), respectively. Likewise, among the 20 *Salmonella* isolates of sonali chicken, the prevalence of *S*. Typhimurium and untyped *Salmonella* spp. were found 55% (11 in 20) and 45% (nine in 20), respectively. Similarly, the prevalence of *S*. Typhimurium and untyped *Salmonella* spp. were found 66.7% (six in nine) and 33.3% (three in nine) among the nine *Salmonella* isolates of native chicken. No *S*. Enteritidis was isolated from sonali and native chickens. The prevalence of *Salmonella* spp. in broiler (8.62%) was significantly higher than sonali (6.89%) and native (3.1%) chicken (*p* < 0.05). Similarly, the prevalence of *Salmonella enterica* serovars in broiler chicken (80%) was significantly higher than sonali (55%) and native (66.66%) chicken (*p* < 0.05).

### 3.2. Phenotypic Resistance Pattern of Salmonella enterica Serovars

AST results revealed that in case of *S*. Typhimurium highest resistance (100%) was recorded to ciprofloxacin and streptomycin followed by 86.66% to tetracycline, nalidixic acid and gentamicin, 66.66% to ampicillin and 40% to amoxicillin–clavulanate in broiler chicken. Whilst in *S*. Enteritidis, the highest resistance was recorded to streptomycin (100%) followed by 80% to ciprofloxacin, tetracycline, and gentamicin, 20% to amikacin, amoxicillin–clavulanate, azithromycin, and sulphamethazaxole-trimethoprim were recorded in broiler chicken. In sonali chicken, *S*. Typhimurium was found resistant 81.81% to streptomycin and tetracycline followed by 72.72% to ciprofloxacin and gentamicin. In native chicken, *S*. Typhimurium was found resistant to tetracycline (100%), ciprofloxacin (66.7%) and 50% to ampicillin and gentamicin (Table 1). The third generation antibiotics (aztreonam, ceftriaxone, cefotaxime and ceftazidime) were found almost sensitive to all isolates. As shown in Table 2, the highest MAR index value of 0.62 was found in one *S*. Typhimurium isolate. The more prevalent MAR index value of 0.43 was recorded in 14 *S*. Typhimurium and two *S*. Enteritidis isolates. In the present study, 78.1% (25 in 32) isolates of *S*. Typhimurium and 80% (four in five) isolates of *S*. Enteritidis were found MDR. All MDR isolates were resistant to at least four of the antimicrobials. A single *S*. Typhimurium isolate was found sensitive to all antimicrobials. The phenotypic AMR pattern was significantly higher in broiler compared to sonali and native chicken (*p* < 0.05). The AMR profile of *Salmonella enterica* serovars are presented in Table 1 and Table 2. The results anticipate that antimicrobials are widely used in broiler production followed by sonali chicken production. The detail AST result is represented in Appendix A.

### 3.3. Genotypic Resistance Pattern

All isolates were screened for seven ESBL producing genes (*bla*TEM, *bla*SHV, *bla*OXA, *bla*CTX-M-1, *bla*CTX-M-2, *bla*CTX-M-9 and *bla*CTX-Mg8/25). Only one ESBL gene, *bla*TEM has been detected and the prevalence among the *S*. Typhimurium isolates were found 73.3%, 63.6% and 50% in broiler, sonali and native chicken, respectively. Moreover, all isolates were examined for three tetracycline resistance genes (*tet*A, *tet*B, and *tet*C) but only one *tet*A gene was detected. The prevalence of *tet*A gene among the *S*. Typhimurium isolates were found 80%, 90.9%, and 100% in broiler, sonali, and native chicken, respectively. Similarly, only *sul*1 gene was encountered out of three resistance genes (*sul*1, *sul*2 and *sul*3) with prevalence rate 80%, 36.4%, and 66.7% among the *S*. Typhimurium isolates of broiler, sonali and native chicken, respectively. In addition, streptomycin resistance gene *str*A/B was detected in few isolates of *S*. Typhimurium with prevalence rate 33.3%, 27.3%, and 16.7% in broiler, sonali and native chicken, respectively. Moreover, the prevalence of *bla*TEM, *tet*A, *sul*1 and *str*A/B were found 40%, 100%, 20%, and 20%, respectively among the *S*. Enteritidis isolates in broiler chicken. The *tet*A gene was found more prevalent compared to other genes (*bla*TEM, *sul*1 and *str*A/B) in broiler, sonali and native chicken. The genotypic resistance patterns are presented in Table 3 as well as in Appendix A.

### 3.4. Virulence Characterization 

All *Salmonella enterica* serovars were screened by PCR to monitor eight common virulence genes namely *inv*A, *agf*A, *Ipf*A, *hil*A, *siv*H, *sef*A, *sop*E, and *spv*C. All five isolates of *S*. Enteritidis were found to be positive for altogether eight virulence genes whist *S*. Typhimurium isolates were found positive for six virulence genes except *sef*A and *spv*C. There was a statistically significant correlation exist among the virulence genes in broiler, sonali, and native chicken (*p* < 0.05). The prevalence of virulence genes are presented in Table 4.

## 4. Discussion

Pathogenic MDR *Salmonella enterica* serovars are a leading cause of foodborne diseases and serious public health concern worldwide. *Salmonella* induced foodborne illness has got more priority than have other foodborne pathogens worldwide. *Salmonella enterica* is highly diverse, having over 2500 different serovars distributed across the globe. In the present study, the overall prevalence rate of *Salmonella* spp., was recorded 6.2% in chicken cecal contents which is very alike with the findings of Mir et al. [67] and Mir et al. [68] who reported an overall prevalence of 6.88% in Kashmir Valley and 6.31% in Rajasthan, India. However, the prevalence rate was lower compared to other studies conducted in Bangladesh, India, Ethiopia and Malaysia [9,69,70]. The low prevalence could be due to the collection of cecal content from healthy chickens as well as precautions were taken to avoid cross contamination during collection of samples at wet markets. The isolation of *S.* Typhimurium and *S*. Enteritidis among the *Salmonella* isolates recovered from different chicken cecal contents are in agreement with other findings across the globe. In Bangladesh, *S*. Typhimurium were isolated from commercial broiler and breeder farms [9,10,71]. In Malaysia, *S*. Typhimurium and *S*. Enteritidis were isolated from raw chicken meat at retail markets [70]. In Turkey, the prevalent serotype was identified as *S*. Enteritidis (21.9%) and *S*. Typhimurium (9.4%) from chicken [72]. In India, *Salmonella enterica* serovars were isolated from backyard poultry flocks [73]. *S*. Enteritidis and *S*. Typhimurium recovered from chicken meat in Egypt [74]. Similar observations had been reported by Suresh et al. [11,75] who recovered *S*. Typhimurium and *S*. Enteritidis in high proportion compared to other serovars from various poultry products in India. *S*. Enteritidis and *S*. Typhimurium were the most prevalent serotypes, consistent with earlier reports from China and some European countries [76,77]. The serovars identified in this study indicate the diversity of *Salmonella* spp. in commercial as well as native poultry flocks in Bangladesh.

*S*. Typhimurium and *S*. Enteritidis have been emerged as major cause of foodborne salmonellosis over the last few decades worldwide [78,79]. *S*. Enteritidis and *S*. Typhimurium are the most predominant isolates in most *Salmonella* cases associated with the consumption of contaminated poultry, pork, and beef products [80,81]. Contamination of *Salmonella* serovars in poultry products can occur at multiple steps along the food chain, including production, processing, distribution, retail marketing, handling, and preparation [82]. *S*. Typhimurium, with its broader range of host tropism, is one of the top two serovars responsible for causing infections in human and animal worldwide [83,84].

The high resistance of *S*. Enteritidis and *S*. Typhimurium to ciprofloxacin, streptomycin, gentamicin, ampicillin, tetracycline, and nalidixic acid in our study predict that these antibiotics are widely used in the poultry farming system in Bangladesh. These findings are in agreement with Alam et al. [9] who reported a range of 77.1% to 97.1% resistance of *Salmonella* isolates to tetracycline, ampicillin, streptomycin, and chloramphenicol. Consequently, Bupasha et al. [85] reported *Salmonella* recovered from pigeons were resistance at the rate of 93.1%, 81.8%, and 86.2% to amoxicillin, erythromycin, and tetracycline, respectively. Suresh et al. [11] found 52.3% and 38.1% resistance to ampicillin and tetracycline in the *Salmonella* isolates of animal origin. The level of resistance against nalidixic acid is very much in agreement with the findings of Halimi et al. [86] who found 53% resistance to nalidixic acid. The resistance patterns are also incompatible and supported with other preceding findings of home and abroad [87]. *Salmonella* isolated from fecal samples of domestic animals (chickens, ducks, geese and pigs) were resistant to nalidixic acid (48.8%), tetracycline (46.9%), sulfafurazole (45.7%), ampicillin (43.2%), streptomycin (38.3%) and trimethoprim/sulfamethoxazole (33.3%) [32,75,88,89]. In our study, MDR *Salmonella* embedded with mostly ciprofloxacin, ampicillin, tetracycline, streptomycin and gentamicin as these drugs are commonly used in poultry production cycle in Bangladesh [90,91]. It is a matter of fact that following the guideline of CLSI [53], the first and second generations of cephalosporins and aminoglycosides may appear active in vitro, but are not effective clinically and must not be reported as susceptible, i.e., *Salmonella* becomes natural resistant to first- and second-generation cephalosporin and aminoglycosides. It is alarming that watch group antibiotic ciprofloxacin had become highly resistant and azithromycin is becoming resistant, though the WHO noted that the watch group have higher resistance potential and are recommended as essential first or second choices in empiric treatment options for a limited number of specific infectious syndromes [92]. This evidence reflects the insight of indiscriminate use of antimicrobial in farming system.

Beta lactum resistant *bla*TEM gene were found 73.3%, 63.6%, and 50% in *S*. Typhimurium isolates of broiler, sonali and native chicken, respectively. The prevalence of *tet*A gene among the *S*. Typhimurium isolates were found 80%, 90.9%, and 100% in broiler, sonali and native chicken, respectively. Similarly, only *sul*1 gene was encountered out of three resistance genes (*sul*1, *sul*2, and *sul*3) with prevalence rate 80%, 36.4%, and 66.7% in *S*. Typhimurium isolates of broiler, sonali and native chicken, respectively. The streptomycin resistance gene, *str*A/B was found 33.3%, 27.3%, and 16.7% in broiler, sonali and native chicken among the *S*. Typhimurium isolates. In our study, there was a harmonic correlation between genotypic and phenotypic resistance decoration. These findings are in agreement and in accordance with the findings of former study conducted at home and abroad [9,54,93]. However, some disagreement was observed between phenotypic and genotypic resistance pattern of sulfamethoxazole-trimethoprim. The causes of disagreement may be due to misalignment of disk (sulfamethoxazole-trimethoprim) and primers (sulfonamide). Moreover, the disagreement between phenotypic and genotypic results may happen due to sensitivity and specificity of disk, primers, concentration of inoculum, laboratory capacity and individual skill. Rather, some research findings supported the misalignment between genotypic and phenotypic resistance pattern [94,95]. Ahmed et al. [96] reported higher prevalence of *bla*TEM gene mediated ESBL production among *Salmonellae* isolated from humans in Bangladesh. Yang et al. [97] detected *bla*TEM gene, a gene encoded for beta-lactamases resistance, in 51.6% resistant *Salmonella* isolates. Aslam et al. [98] reported that the percentage of *bla*TEM gene in *Salmonella* isolated from retail meats in Canada was 17% and this gene was the most common resistance genes detected. Lu et al. [99] observed that 81.2% *bla*TEM gene, while *bla*CTX-M could not be detected in any of the examined isolates. Similarly, Van et al. [94] found only *bla*TEM gene in *E. coli* recovered from raw meat and shellfish in Vietnam. The emergence of *bla*TEM mediated ESBL producing *Salmonella enterica* serovars indicates the use of beta-lactam antibiotics in poultry farming practices. Moreover, ESBL are usually encoded by large plasmids that are transferable from strain to strain and between bacterial species [96,100]. Arkali and Çetinkaya [72] detected *sul1* gene with 58% among the *Salmonella* isolates from chickens in eastern Turkey. Consequently, Jahantigh et al. [101] isolated the most prevalent *tet*A from broiler chicken in Iran. Accordingly, Vuthy et al. [89] detected *bla*TEM, *tet*A, *str*A/B gene from chicken food chain while Sin et al. [102] isolated *tet*A and *sul*1 gene from chicken meat in Korea. Our molecular detection of resistant gene is consistent with many findings across the globe including Bangladesh. In contrast, other ESBL genes (*bla*SHV, *bla*OXA, *bla*CTX-M), tetracycline genes (*tet*B, *tet*C), and sulfonamide genes (*sul*2, *sul*3) were not detected in any isolates in this study.

The presence of virulence genes and AMR pattern can accelerate the pathogenicity of the microbes [20]. The emergence of AMR of *Salmonella enterica* solely depends on genetic and pathogenicity mechanisms that may enhance the survivability by preserving their drug resistance genes [98]. The virulence gene was more prevalent in *S*. Enteritidis compared to *S*. Typhimurium isolates. The virulence genes *inv*A, *agf*A, *Ipf*A, *hil*A, *siv*H, and *spv*C were detected in all isolates of *Salmonella enterica* which is consistent with earlier findings across the world [103,104,105,106]. Furthermore, the virulence gene *sef*A and *spv*C were detected only in *S*. Enteritidis isolates and on the contrary no *sef*A and *spv*C genes were detected in any *S*. Typhimurium isolates. Similar observations have been recorded previously by researchers [104]. The high prevalence of *sef*A in *S*. Enteritidis is interlinked with prior findings [103,106], and *sef*A is recognized a target gene to detect *S*. Enteritidis serovars in molecular method [103]. The *inv*A is the most virulent and common gene present in *Salmonella* which is target gene to identify *Salmonella* spp. [18,81,107]. The *hil*A gene play a key role in *Salmonella* virulence through stimulate the expression of invasion [108,109]. The virulence gene *inv*A and *hil*A can be considered target genes to rapid detection of *Salmonella* spp. through PCR method. The higher incidence of *lpf*A and *agf*A were comparable to previous research findings on different serovars [23,110]. The frequency of *sop*E gene (100%) were alike with earlier studies on *S*. Enteritidis [111]. Further, *agf*A gene responsible with biofilm formation as well as adhesion during infection process [112]. In our result, the virulence plasmid gene (*spv*C) was only detected in *S*. Enteritidis which has similarities with earlier observations [103,113,114]. It has been observed previously that 92% of *S*. Enteritidis strains had the *spv*C gene, whereas only 28% to *S*. Typhimurium and no gene found in *S*. Hadar [115]. It was found that *S*. Enteritidis and *S*. Typhimurium exposed to wider range of pathogenicity compared to other serovars. The presence of important virulence gene reflects the pathobiology as well as public health significance of the serovars. Thus, all the *Salmonella enterica* isolates were found highly invasive and enterotoxigenic which had great public health impact. This is the first attempt to encounter a wider range of virulence gene of *Salmonella enterica* serovars in Bangladesh.

Our findings demonstrate that MDR strains of *S*. Enteritidis and *S*. Typhimurium are prevalent in the cecal contents of broiler, sonali and native chicken. In reality, MDR *Salmonella* serovars are denoted more virulent than non-MDR *Salmonella* [18]. This might lead to human infections with foodborne AMR *Salmonella*, and probably create an enormous challenge to treatment of *Salmonella* infection in humans and animals in Bangladesh. Various reports on the risk factor associated with the occurrence of MDR *Salmonella* isolates have been published. Ziech et al. [34] reported that the appearance of MDR *Salmonella* isolates correlates positively with the indiscriminate use of antibiotics at recommended doses or at sub-therapeutic doses as feed additives in poultry farm. In addition, genetic and biochemical mechanisms may make a significant contribution to the emergence of MDR strains of *Salmonella*, and thus preserve their drug resistance genes and enhance their survivability.

Our results also indicate that wet markets where chicken are processed act as reservoirs in harboring *Salmonella enterica* serovars. In wet markets, cross-contamination might occur during the dressing and processing of chicken due to poor sanitary and hygienic measures. The chicken carcass may be contaminated with MDR *Salmonella* serovars due to cross contamination during slaughter and facilitate the dissemination of the resistance genes to consumers along the production chain, which suggests importance of controlling *Salmonella* during slaughter. Moreover, the recovered MDR *Salmonella enterica* serovars constitute a possible risk to human. Therefore, it is important to manage the use of antimicrobial agents in poultry farming system to prevent the acquisition and increased resistance to recent molecules in order to fight against the vertical and horizontal transfer of MDR strains. Alternatively, it is necessary to develop more effective intervention strategies, such as sanitation, drainage, waste management, awareness, and training, in order to reduce the risk of foodborne diseases at wet markets. The study has raised a serious public health concern and thus demands strict monitoring and surveillance at wet markets.

## 5. Conclusions

The detection of pathogenic MDR *Salmonella enterica* serovars Typhimurium and Enteritidis from cecal contents of healthy chickens at retail wet markets remains extremely alarming and has led to great public health concern. Commercial and native both chickens are carriers of pathogenic MDR *Salmonella enterica* serovars. The result depicted that *Salmonella enterica* serovars harbor at wet markets where chicken carcass is processed. The wet market could be considered the hot spot of spread and contamination of MDR *Salmonella enterica* serovars which can easily anchor to vendor, consumers and in food chain due to poor sanitary and unhygienic measures. Moreover, the emergence of resistance in healthy chickens may be generated due to the irrational use of antimicrobials in the production cycle or spill over from the environment. Results of this study would help to effective designing and implementation of national AMR surveillance strategy for ensuring food safety and market management to further minimize the spread at wet markets. Further resistance patterns would impart a message to physicians, researchers, and policy makers to formulate standard treatment guidelines and support the adoption of good agricultural practices for the prudent and judicious use of antimicrobials in the poultry production system.

## Figures and Tables

**Table 1 microorganisms-09-00952-t001:** AMR patterns of *Salmonella enterica* serovars in broiler, sonali and native chicken.

Antimicrobials	Antibiogram				Level of Significance
	Broiler		Sonali	Native	
	*S*. Typhimurium % (*n*/*N*)	*S*. Enteritidis % (*n*/*N*)	*S*. Typhimurium % (*n*/*N*)	*S.* Typhimurium % (*n*/*N*)	
Ciprofloxacin	100 (15/15)	80 (4/5)	72.7 (8/11)	66.7 (4/6)	**
Streptomycin	100 (15/15)	100 (5/5)	72.7 (8/11)	33.3 (2/6)	**
Ampicillin	66.7 (10/15)	60 (3/5)	72.7 (8/11)	50 (3/6)	**
Tetracycline	86.7 (13/15)	80 (4/5)	72.7 (8/11)	100 (6/6)	**
Nalidixic acid	86.7 (13/15)	60 (3/5)	45.5 (5/11)	33.3 (2/6)	**
Gentamicin	86.7 (13/15)	80 (4/5)	72.7 (8/11)	50 (3/6)	**
Azithromycin	13.3 (2/15)	20 (1/5)	9.1 (1/11)	0	ns
amoxicillin–clavulanate	40 (6/15)	20 (1/5)	27.3 (3/11)	0	ns
Chloramphenicol	6.7 (1/15)	0	0	16.7 (1/6)	ns
Sulphamethazaxole-Trimethoprim	0	20 (1/5)	0	16.7 (1/6)	ns
Amikacin	13.3 (2/15)	20 (1/5)	9.1 (1/11)	0	ns
Meropenem	0	0	9.1 (1/11)	16.7 (1/6)	ns
ceftazidime	13.3 (2/15)	20 (1/5)	9.1 (1/11)	0	ns
Ceftriaxone	13.3 (2/15)	20 (1/5)	0	0	ns
Cefotaxime	13.3 (2/15)	0	0	0	ns
Aztreonam	6.7 (1/15)	0	0	0	ns

** = significant (*p* < 0.05), ns = non-significant, *n* = Number of resistant isolate, *N* = Number of *Salmonella* isolates.

**Table 2 microorganisms-09-00952-t002:** AMR patterns and MAR index of *S.* Typhimurium and *S.* Enteritidis isolated from chicken at wet markets.

Isolate No.	Sources	Serovars	Resistance Profile	^1^ MAR Index
ARAC-CD-CH-1510	BC	*S.* Typhimurium	CIP-S-AMP-TE-NA-CN-AZM	0.43
ARAC-CD-CH-1687	BC	*S.* Typhimurium	CIP-S-AMP-TE-NA-CN-AMC	0.43
ARAC-CD-CH-1861	BC	*S.* Typhimurium	CIP-S	0.12
ARAC-CD-CH-1929	BC	*S.* Typhimurium	CIP-S-AMP-TE-NA-CN-CAZ	0.43
ARAC-CD-CH-2441	BC	S. Typhimurium	CIP-S-AMP-TE-NA-CN-AMC	0.43
ARAC-CD-CH-2503	BC	*S.* Typhimurium	CIP-S-AMP-TE-NA-CN-AMC	0.43
ARAC-CD-CH-2564	BC	*S.* Typhimurium	CIP-S-AMP-TE-NA-CN-AMC	0.43
ARAC-CD-CH-2628	BC	*S.* Typhimurium	CIP-S-AMP-TE-NA-CN-CRO	0.43
ARAC-CD-CH-2688	BC	*S.* Typhimurium	S-AZM	0.12
ARAC-CD-CH-2750	BC	*S.* Typhimurium	CIP-S-AMP-TE-NA-CN	0.37
ARAC-CD-CH-2801	BC	*S.* Typhimurium	CIP-S-AMP-TE-NA-CN-AMC	0.43
ARAC-CD-CH-2803	BC	*S.* Typhimurium	CIP-S-AMP-TE-NA-CN-AMC	0.43
ARAC-CD-CH-2806	BC	*S*. Typhimurium	CIP-S-AMP-TE-NA-CN	0.38
ARAC-CD-CH-2987	BC	*S.* Typhimurium	CIP-S-TE-NA-AMC-C-AK-CT-CRO ATM	0.62
ARAC-CD-CH-3044	BC	*S.* Typhimurium	CIP-S-TE-NA-CN	0.31
ARAC-CD-CH-1519	SC	*S*. Typhimurium	CIP-AMP-TE-NA-MEM	0.31
ARAC-CD-CH-1576	SC	*S*. Typhimurium	CIP-S-AMP-TE-NA-CN-AMC	0.43
ARAC-CD-CH-1991	SC	*S*. Typhimurium	S-AMP-TE-NA-CN-AK	0.37
ARAC-CD-CH-2573	SC	*S*. Typhimurium	CIP-S-AMP-TE-NA-CN-AMC-AK	0.5
ARAC-CD-CH-2511	SC	*S*. Typhimurium	0	0
ARAC-CD-CH-2573	SC	*S.* Typhimurium	CIP-S-AMP-TE-NA-CN-AMC	0.43
ARAC-CD-CH-2633	SC	*S.* Typhimurium	S-AZM	0.12
ARAC-CD-CH-2756	SC	*S*. Typhimurium	CIP-S-AMP-TE-CN	0.31
ARAC-CD-CH-2813	SC	*S.* Typhimurium	CIP-S-AMP-TE-NA-CN-AMC-CT	0.5
ARAC-CD-CH-3115	SC	*S*. Typhimurium	CIP-S-TE-CN	0.25
ARAC-CD-CH-3173	SC	*S*. Typhimurium	CIP-S-AMP-TE-NA-CN-CT	0.43
ARAC-CD-CH-1523	NC	*S.* Typhimurium	CIP-AMP-TE-NA-MEM	0.31
ARAC-CD-CH-2881	NC	*S*. Typhimurium	CIP-S-AMP-TE-CN-SXT-C	0.43
ARAC-CD-CH-3004	NC	*S*. Typhimurium	CIP-TE	0.12
ARAC-CD-CH-3130	NC	*S*. Typhimurium	S-TE	0.12
ARAC-CD-CH-3185	NC	*S.* Typhimurium	TE-CN	0.12
ARAC-CD-CH-3186	NC	S. Typhimurium	CIP-S-AMP-TE-NA-CN-AMC	0.43
ARAC-CD-CH-1929	BC	*S*. Enteritidis	CIP-S-AMP-TE-NA-CN-CAZ	0.43
ARAC-CD-CH-1988	BC	*S*. Enteritidis	CIP-S-TE-CN-AMC-SXT	0.37
ARAC-CD-CH-2684	BC	*S*. Enteritidis	S	0.06
ARAC-CD-CH-2686	BC	*S*. Enteritidis	CIP-S-AMP-TE-NA-CN-AK-CRO	0.5
ARAC-CD-CH-2747	BC	*S*. Enteritidis	CIP-S-AMP-TE-NA-CN-AZM	0.43

BC = Broiler ceca, SC = Sonali ceca, NC = Native ceca, ^1^ MAR index = number of resistance antibiotics/total number of antibiotics tested. AK-amikacin, gentamicin (CN), streptomycin (S), meropenem (MEM); ceftriaxone (CRO), cefotaxime (CT), ceftazidime (CAZ), aztreonam (ATM); amoxicillin–clavulanate (AMC), ampicillin (AMP), azithromycin (AZM), ciprofloxacin (CIP), nalidixic acid (NA), sulfamethoxazole-trimethoprim (SXT), tetracycline: tetracycline (TE), chloramphenicol (C).

**Table 3 microorganisms-09-00952-t003:** Prevalence of AMR gene in *S*. Typhimurium and *S*. Enteritidis in different chicken.

Samples	Serovars	*bla*TEM %	*Tet*A %	*Sul*1 %	*Str*A/B %
Broiler ceca	*S.* Typhimurium	73.3 (11/15)	80 (12/15)	80 (12/15)	33.3 (5/15)
	*S.* Enteritidis	40 (2/5)	100 (5/5)	20 (1/5)	20 (1/5)
Sonali ceca	*S.* Typhimurium	63.6 (7/11)	90.9 (10/11)	36.4 (4/11)	27.3 (3/11)
Native ceca	*S.* Typhimurium	50 (3/6)	100 (6/6)	66.7 (4/6)	16.7 (1/6)

**Table 4 microorganisms-09-00952-t004:** Prevalence of virulence genes of *Salmonella enterica* serovar.

Samples	Serovars	*Inv*A %	*Agf*A %	*Ipf*A %	*Hil*A %	*Siv*H %	*Sef*A %	*Sop*E %	*Spv*C %
Broiler ceca	*S.* Typhimurium	100 (15/15)	100 (15/15)	100 (15/15)	100 (15/15)	100 (15/15)	0	100 (15/15)	0
	*S.* Enteritidis	100 (5/5)	100 (5/5)	100 (5/5)	100 (5/5)	100 (5/5)	100 (5/5)	100 (5/5)	100 (5/5)
Sonali ceca	*S.* Typhimurium	100 (11/11)	100 (11/11)	100 (11/11)	100 (11/11)	100 (11/11)	0	100 (11/11)	0
Native ceca	*S.* Typhimurium	100 (6/6)	100 (6/6)	100 (6/6)	100 (6/6)	100 (6/6)	0	100 (6/6)	0
Level of significance		**	**	**	**	**	ns	**	ns

** significant (*p* < 0.05), ns-non significant.

## Data Availability

The data presented in this study are available either in the article or Appendix A here.

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
