# Peer review of "Virulence and Antimicrobial Resistance Profiles of Salmonella enterica Serovars Isolated from Chicken at Wet Markets in Dhaka, Bangladesh"

_microorganisms, 2021, doi:10.3390/microorganisms9050952_

Round 1

Reviewer 1 Report

In this manuscript, Siddiky and colleagues report the genetic and phenotypic characterization of Salmonella enterica serovars isolated from chicken at wet markets in Dhaka, Bangladesh. This paper will be an important contribution to Microorganisms if major concerns are addressed.

Major concerns:

  1. Phenotypic and genotypic differences in antibiotic-resistant patterns were observed, which deserves extended explanations and further discussion. For instance, what control experimental data support the integrity of phenotypic profiles and genotypic profiles. Suppose the additional data support that obtained results are not due to some particular sample running issues. In that case, the discussion should be extended to explain why the phenotype-genotype disagreement was observed with these Salmonella isolates.

  1. Only the part of the isolated Salmonella samples was further characterized for phenotypic and genotypic drug resistance profiles. The selection strategy (relating to what criteria were used to select only a part from all Salmonella samples for further analysis) is not indicated in the manuscript.

  1. The meaning of the term MDR in this manuscript should be defined in the beginning. This is important to avoid confusion since MDR Salmonella is typically referred to as the strain resistant to first-line antibiotics, ampicillin, chloramphenicol, and sulphamethoxazole-trimethoprim. It is interesting to see that none of the Salmonella isolates from chicken are equipped with conventional “MDR” phenotypes and genotypes.

  1. Even the typical “MDR” profiles are not observed, Salmonella isolates’ resistance to ciprofloxacin is notable. Is Ciprofloxacin regularly treated to these food animals in Bangladesh during farming? The authors’ insight into this phenotype could be extended in the Discussion.

  1. Among the seven ESBL genes analyzed, only one, blaTEM, has been detected. Authors’ insights into why this is the case will be beneficial to this manuscript. This finding can be discussed further in the context of similarities and differences with relevant isolates from people and other animals in the region and isolates from the different areas.

  1. Some of the results should reflect these profiles at an individual isolate level to understand their relationships between phenotypic and genotypic drug-resistant profiles. Presenting bulk data (percentage) is not so informative.

Minor concerns:

Lines 283-285: It is unclear what correlation was analyzed to be significant

Author Response

Reviewer concerns/ suggestions

Actions/ Interventions

Remarks

Phenotypic and genotypic differences in antibiotic-resistant patterns were observed, which deserves extended explanations and further discussion. For instance, what control experimental data support the integrity of phenotypic profiles and genotypic profiles. Suppose the additional data support that obtained results are not due to some particular sample running issues. In that case, the discussion should be extended to explain why the phenotype-genotype disagreement was observed

In our study we found good correlation between phenotypic and genotypic result of ampicillin, tetracycline and streptomycin (strA/B) but it was observed few disagreement in case of sulfamethoxazole-trimethoprim. In case of sulfonamide we used antibiotic disk sulfamethoxazole-trimethoprim for phenotypic and used sulfonamide gene (sul1, sul2, sul3) for genotypic purposes. The causes of disagreement may be due to misalignment of disk and primers. Moreover, the disagreement between phenotypic and genotypic results may happen due to sensitivity and specificity of disk, primers, concentration of inoculum, laboratory capacity and individual skill.

Rather, some research findings supported the misalignment between genotypic and phenotypic resistance pattern (Van et al., 2008; Paterson et al., 2005). 

The justification has given in  line numbers

373-379

Only the part of the isolated Salmonella samples was further characterized for phenotypic and genotypic drug resistance profiles. The selection strategy (relating to what criteria were used to select only a part from all Salmonella samples for further analysis) is not indicated in the manuscript.

In our manuscript we wanted to focus the antimicrobial resistance profile of Salmonella enterica serovars, i.e. S. Enteritidis and S. Typhimurium due to their public health importance. So, considering public health importance we have characterized a part of Salmonella for phenotypic and genotypic drug resistance profile.  

The justification is stated in text. See line lumber 103-104

The meaning of the term MDR in this manuscript should be defined in the beginning. This is important to avoid confusion since MDR Salmonella is typically referred to as the strain resistant to first-line antibiotics, ampicillin, chloramphenicol, and sulphamethoxazole-trimethoprim. It is interesting to see that none of the Salmonella isolates from chicken are equipped with conventional “MDR” phenotypes and genotypes.

MDR stands for multi drug resistant.

As you rightly pointed out that ampicillin, chloramphenicol and   sulphamethoxazole-trimethoprim may consider first line antibiotics to be embedded for MDR. But in our country context, chloramphenicol and sulphamethoxazole-trimethoprim are not commonly used in poultry production system. Therefore, most MDR Salmonella embedded with ciprofloxacin, ampicillin, tetracycline, streptomycin as these drugs are mostly used in poultry production cycle.

The abbreviation of MDR has included in the text (line number-17

MDR embedded with antibiotics are stated in lines 351-353

Even the typical “MDR” profiles are not observed, Salmonella isolates’ resistance to ciprofloxacin is notable. Is Ciprofloxacin regularly treated to these food animals in Bangladesh during farming? The authors’ insight into this phenotype could be extended in the Discussion.

Ciprofloxacin is one of the common antimicrobial which is used very regularly to the food animals in Bangladesh. Different studies in Bangladesh showed ciprofloxacin, tetracycline, gentamicin and ampicillin are commonly used in poultry production system (Al Masud et al., 2020; Parvej et al., 2016). 

It has been reflected in the discussion section (line number 337-339)

Among the seven ESBL genes analyzed, only one, blaTEM, has been detected. Authors’ insights into why this is the case will be beneficial to this manuscript. This finding can be discussed further in the context of similarities and differences with relevant isolates from people and other animals in the region and isolates from the different areas.

Ahmed et al. (2014) reported higher prevalence of blaTEM gene mediated ESBL production among Salmonellae isolated from humans in Bangladesh. Yang et al. (2010) detected blaTEM gene, a gene encoded for beta-lactamases resistance, in 51.6% resistant
Salmonella isolates. Aslam et al. (2012) reported blaTEM as the most common resistance gene in Salmonella isolated from retail meats in Canada. Lu et al. (2011) observed 81.2% blaTEM gene, while blaCTX could not be detected in any of the chicken isolates. The emergence of blaTEM mediated ESBL producing Salmonella enterica serovars indicates the use of beta-lactam antibiotics in poultry farming practices. Van et al. (2008) found only blaTEM gene in E. coli recovered from raw meat and shellfish in Vietnam. Moreover, ESBL are usually encoded by large plasmids that are transferable from strain to strain and between bacterial species.  

This is reflected in the line numbers 379-390

Lines 283-285: It is unclear what correlation was analyzed to be significant

we have revised the statements 

It is stated in line 290-292

Reviewer 2 Report

Salmonella serovars are an important foodborne pathogen with global public health significance and the antibiotic resistance associated with this pathogen is a great concern as well. However, the current manuscript has major deficiencies for publication consideration in this journal with impact factor >4. Specifically:

-References are not cited in order. After 42, reference 92 is cited. Also, the vast majority of the references are very outdated, and very few studies are included from 2020 and 2021. Key references discussing the importance of antibiotic resistance and Salmonella was not discussed, included:

CDC Antimicrobial resistance report: https://www.cdc.gov/drugresistance/pdf/threats-report/2019-ar-threats-report-508.pdf

WHO Antimicrobial resistant report: https://www.who.int/antimicrobial-resistance/interagency-coordination-group/IACG_final_report_EN.pdf?ua=1

In addition, references are not cited carefully, for example, on line 88, authors cite four references but some of the information are not supporting by the references provided. The study of Zhao et al., 2016, as an example, does not support the preceding sentence and is in a different area of research (the study is about cellulose degradation during composting but is cited to support phenotypic and genotypic resistance gene in the food production).

-The statistical analysis section and analytical information are very limited in scope, challenging the reproducibility of the study. As an example, in section 2.1, the authors discuss the calculation of sample size but do not discuss what software was used for the calculation and what statistical power level was used.

Due to shortcomings in references as well and analytical and microbiological sections, I would need to suggest rejection of the study at the current state.

Author Response

Reviewer concerns/ suggestions

Actions/ Interventions

Remarks

-References are not cited in order. After 42, reference 92 is cited. Also, the vast majority of the references are very outdated, and very few studies are included from 2020 and 2021. Key references discussing the importance of antibiotic resistance and Salmonella was not discussed, included:

Reference has been cited in order.  

The very old references has been removed.

The importance of antibiotic resistance has been included following the suggestion of reviewer.

CDC Antimicrobial resistance report: https://www.cdc.gov/drugresistance/pdf/threats-report/2019-ar-threats-report-508.pdf

The key information has taken and incorporated in the document

It is reflected in line number 76-80

WHO Antimicrobial resistant report: https://www.who.int/antimicrobial-resistance/interagency-coordination-group/IACG_final_report_EN.pdf?ua=1

The key information has taken and incorporated in the document

It is reflected in line number 80-84

In addition, references are not cited carefully, for example, on line 88, authors cite four references but some of the information are not supporting by the references provided. The study of Zhao et al., 2016, as an example, does not support the preceding sentence and is in a different area of research (the study is about cellulose degradation during composting but is cited to support phenotypic and genotypic resistance gene in the food production).

The references has been checked very carefully in alignment with statements. Removed irrelevant  and unaligned references from the manuscript.     

Zhao et al., 2016 is removed

The statistical analysis section and analytical information are very limited in scope, challenging the reproducibility of the study. As an example, in section 2.1, the authors discuss the calculation of sample size but do not discuss what software was used for the calculation and what statistical power level was used

In our manuscript we have only analyzed microbial laboratory data. So, really it has limited scope to explore the data statistically with different dimensions. In our manuscript we have only disclosed correlation among the three different poultry species with prevalence, phenotypic, genotypic and virulence patterns.

For calculation of sample size we used “Sample Size Calculator for Prevalence Studies, version 1.0.01. April, 2006”. Available at: http:// www.kck.usm.my/ppsg/stats_resources.htm

It is reflected in page number 120-121 

Round 2

Reviewer 1 Report

The authors have addressed my concerns.

Reviewer 2 Report

Although there has been some improvement for the manuscript during the revision, my previous concerns still remain for this paper. In addition to the limited sample size, the study statistical methodology section is very limited in scope. The statistical analysis section is nearly copied and pasted from the below study that appears to be from the same country. You are welcome to compare the statistical analyses section of this manuscript with the below-mentioned study of Alam et al., 2020. It is nearly identical.

Alam, S.B.; Mahmud, M.; Akter, R.; Hasan, M.; Sobur, A.; Nazir, N.H.; et al. Molecular detection of multidrug-resistant Salmonella 504 species isolated from broiler farm in Bangladesh. Pathogens 2020, 9, 1–12

My assessment would be rejecting the article at the current state due to methodological concerns.